# Using parasite genetic and human mobility data to infer local and cross-border malaria connectivity in Southern Africa

Sofonias Tessema[1†*], Amy Wesolowski[2†], Anna Chen[1], Maxwell Murphy[1], Jordan Wilheim[1], Anna-Rosa Mupiri[3], Nick W Ruktanonchai[4], Victor A Alegana[3,4], Andrew J Tatem[4], Munyaradzi Tambo[3], Bradley Didier[5], Justin M Cohen[5], Adam Bennett[6], Hugh JW Sturrock[6], Roland Gosling[3,6], Michelle S Hsiang[6,7,8], David L Smith[9], Davis R Mumbengegwi[3], Jennifer L Smith[6], Bryan Greenhouse[1,10]

[1]EPPIcenter program, Division of HIV, Infectious Diseases and Global Medicine, Department of Medicine, University of California, San Francisco, San Francisco, United States; [2]Department of Epidemiology, Johns Hopkins Bloomberg School of Public Health, Baltimore, United States; [3]Multidisciplinary Research Center, University of Namibia, Windhoek, Namibia; [4]WorldPop Project, Geography and Environment, University of Southampton, Southampton, United Kingdom; [5]Clinton Health Access Initiative, Boston, United States; [6]Malaria Elimination Initiative, Institute of Global Health Sciences, University of California, San Francisco, San Francisco, United States; [7]Department of Pediatrics, University of Texas Southwestern Medical Center, Dallas, United States; [8]Department of Pediatrics, UCSF Benioff Children's Hospital, San Francisco, United States; [9]Institute for Health Metrics and Evaluation, University of Washington, Seattle, United States; [10]Chan Zuckerberg Biohub, San Francisco, United States

*For correspondence:
SofoniasK.Tessema@ucsf.edu

†These authors contributed equally to this work

Competing interests: The authors declare that no competing interests exist.

**Abstract** Local and cross-border importation remain major challenges to malaria elimination and are difficult to measure using traditional surveillance data. To address this challenge, we systematically collected parasite genetic data and travel history from thousands of malaria cases across northeastern Namibia and estimated human mobility from mobile phone data. We observed strong fine-scale spatial structure in local parasite populations, providing positive evidence that the majority of cases were due to local transmission. This result was largely consistent with estimates from mobile phone and travel history data. However, genetic data identified more detailed and extensive evidence of parasite connectivity over hundreds of kilometers than the other data, within Namibia and across the Angolan and Zambian borders. Our results provide a framework for incorporating genetic data into malaria surveillance and provide evidence that both strengthening of local interventions and regional coordination are likely necessary to eliminate malaria in this region of Southern Africa.
DOI: https://doi.org/10.7554/eLife.43510.001

# Introduction

Renewed efforts against malaria have resulted in substantial gains in malaria control, with active plans to eliminate malaria from 35 countries (*Newby et al., 2016*). Malaria elimination requires that national and regional strategies consider the impact of local and cross-border importation on local

**eLife digest :** The number of malaria cases has dropped in some Southern Africa countries, but others still remain seriously affected. When people travel within and between countries, they can bring the parasites that cause the disease to different areas. This can fuel local transmission or even lead to outbreaks in a malaria-free area.

When new malaria patients are diagnosed, they are often asked to report their recent travel history, so that the origin of their infection can be tracked. In theory, this would help to spot regions where the disease is imported from, and design targeted interventions.

However, it is difficult to know exactly where the parasites come from based on self-disclosed travel history. At best, this history can provide information about that persons infection but nothing further in the past; at worst this history can be completely incorrect. Parasite DNA, on the other hand, has the potential to bring with it an indelible record of the past. To address the problem of determining where malaria infections came from, Tessema, Wesolowski et al. focused on Northern Namibia, a region where malaria persists despite being practically absent from the rest of the country. Patients movements were assessed using mobile phone call records as well as self-reported travel history In addition, samples a single drop of blood were taken so that the genetic information of the parasites could be examined.

Combining genetic data with travel history and phone records, Tessema, Wesolowski et al. found that, in Northern Namibia, most people had gotten infected by malaria locally. However, the genetic analyses also revealed that certain infections came from places across the Angolan and Zambian borders, information that was much more difficult to obtain using self-report or mobile phone data. A new, separate study by Chang et al. also supports these results, showing that, in Bangladesh, combining genetic data with travel history and mobile phone records helps to track how malaria spreads.

Overall, the work by Tessema, Wesolowski et al. indicate that, in Northern Namibia, it will be necessary to strengthen local interventions to eliminate malaria. However, different countries in the region may also need to coordinate to decrease malaria nearby and reduce the number of cases coming into the country. While genetic data can help to monitor how new malaria cases are imported, this knowledge will be most valuable if it is routinely collected across countries. New tools will also be required to translate genetic data into information that can easily be used for control and elimination programs.

DOI: https://doi.org/10.7554/eLife.43510.002

transmission (*Cotter et al., 2013*; *Marshall et al., 2016a*; *Wangdi et al., 2015*; *WHO, 2017*). This is particularly important for eliminating countries that share porous borders with areas of higher transmission, where importation can play a major role in sustaining or reestablishing local transmission (*Sturrock et al., 2015*). Identifying within-country and cross-border blocks of high parasite connectivity and coordinating elimination strategies accordingly will likely be required for national and regional success.

Coordinating the optimal interventions to deploy when and where depends on understanding the impact of imported malaria infections on local transmission. If transmission is self-sustained locally, local control measures such as vector control will be necessary. If importation strongly connects the local parasite population to an external one, then interventions aimed at reducing malaria in these sources of importation or otherwise reducing vulnerability to importation may additionally be needed or even take precedence (*Cotter et al., 2013*).

Currently, the extent of importation is estimated primarily by taking recent travel histories of malaria cases (*Sturrock et al., 2015*) and, less commonly, from more general estimates of human mobility. However, routine collection of travel data is not universal, even in areas nearing elimination, and requires a robust surveillance system. When these data are collected, they are often incomplete (e.g. only the most recent travel is reported), or are otherwise inaccurate (e.g. due to disincentives such as reduced access to free healthcare when a patient reports foreign nationality) (*Marshall et al., 2016b*; *Pindolia et al., 2012*). Even with an accurate travel history, it can be difficult to tell with confidence whether malaria parasites were acquired locally or during travel. Beyond

these factors, obtaining travel history only from those presenting with symptomatic malaria does not address the contribution of asymptomatic carriers, which may be the population primarily responsible for importation. Thus, travel data alone are often unable to accurately capture the relative contribution of malaria importation to local transmission. Since travel data are often limited, approaches based on movement of the overall human population using anonymized mobile phone data have been developed to create a generalizable and scalable framework for estimating movement of malaria parasites. However, these methods rely heavily on modeling assumptions, assume that the movement patterns of mobile phone owners and the at risk population are similar, and have not been used to measure international travel (*Pindolia et al., 2012*; *Ruktanonchai et al., 2016*; *Tatem, 2014*; *Tatem et al., 2014*; *Wesolowski et al., 2012*; *Zhao et al., 2016*).

Since travel history and other data on human movement are limited in their ability to provide reliable information on malaria parasite connectivity, it may be valuable for control programs to additionally collect data on parasite genetics (*Wesolowski et al., 2018*). In principle, data on the genetics of malaria parasites have the potential to provide the most direct measure of parasite connectivity and to identify relevant sources and sinks of parasite movement (*Auburn and Barry, 2017*; *Escalante et al., 2015*). However, there have been few efforts to systematically collect and genotype malaria infected individuals at sufficient spatial and temporal scale or density to be useful in this regard. In addition, it has been difficult to detect relevant spatial signals in parasite genetic data using existing population genetic methods, particularly in areas such as sub-Saharan Africa that have high levels of population diversity and polyclonal infections (*Anderson et al., 2000*; *Mobegi et al., 2012*). Ideally, multiple complementary sources of human and parasite data would be compared and integrated to better understand the movement of malaria parasites and contribution to transmission at various spatial scales.

As part of the Elimination 8 (E8) initiative, a regional effort to eliminate malaria from Southern Africa, Namibia has been successful in decreasing malarial morbidity and mortality (*Elimination 8, 2015*). However, this success has recently stalled, with the number of reported cases increasing in the last few years (*Nghipumbwa et al., 2018*; *WHO, 2017*). To achieve the national malaria elimination target of 2020, it will be critical to reassess the elimination strategy in northern Namibia, where nearly all cases in the country are reported. Of particular concern is determining the contribution and spatial scale of local transmission to malaria within northern Namibia, which should guide the geographic coverage and relative timing of local interventions, and the contribution of importation from neighboring Angola and Zambia, which should guide cross-border strategies. To address these concerns, we systematically collected parasite genetic data and human mobility data – travel history from confirmed malaria cases and national mobile phone call data records – from the region in 2015–16. From these data, we aimed to determine the importance of local transmission and imported malaria, and to compare estimates of parasite connectivity at various spatial scales obtained from the different data sources. Our results demonstrate strong evidence for local transmission in northern Namibia, provide insight into patterns of parasite connectivity within Namibia and across national borders, and demonstrate the feasibility of efficiently generating actionable information for malaria control by augmenting traditional surveillance data with a direct evaluation of the parasite population.

## Results

A total of 4643 RDT confirmed, symptomatic malaria cases were enrolled from 29 health facilities in northeastern Namibia; 23 from Kavango East and six from Zambezi regions. The Kavango East survey was conducted between March and June 2016; whereas the Zambezi study was conducted between February 2015 and June 2016 (*Figure 1*, *Figure 1—figure supplement 1*). A subset of these infections (n = 2585, Data found in *Supplementary file 1*) were successfully genotyped from used rapid diagnostic tests (RDTs, n = 2128, Kavango East) and dried blood spots (DBS, n = 457, Zambezi). These data were analyzed along with concomitantly collected travel survey data in these patients and mobile phone data collected from subscribers in the study area (*Ruktanonchai et al., 2016*).

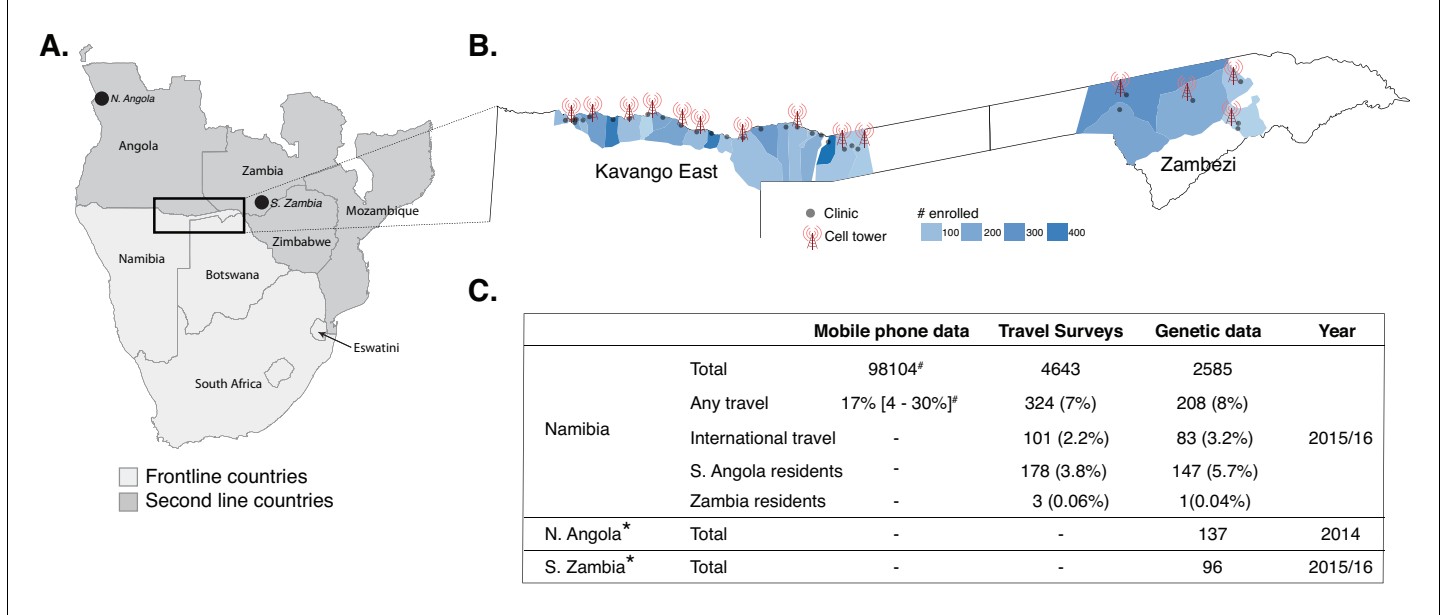

**Figure 1.** Study area and summary of data analyzed. (A) Samples were analyzed from Namibia, northern Angola and southern Zambia (black points and box) from the Elimination 8 (E8) region, which is operationally divided into frontline, low-transmission areas (light gray) and second line, higher transmission areas (dark gray). (B) In Namibia, malaria cases from 29 health facilities in two regions (Kavango East and Zambezi) were enrolled and genotyping data generated on a subset. The locations of the health facilities are shown in gray dots and the sample sizes are shown in blue for the catchment areas of each health facility (*Alegana et al., 2016*). Data from mobile phone subscribers at 14 cell towers in the study area were used to estimate population mobility. (C) Summary of mobile phone, travel survey, and genetic data analyzed. *Additional genotyping data from Northern Angola and Southern Zambia were included in the analyses. #Number of mobile phone subscribers in the study area and percent time spent outside of the primary cell tower.

DOI: https://doi.org/10.7554/eLife.43510.003

The following figure supplement is available for figure 1:

**Figure supplement 1.** Flow chart of samples collected and genotyped.

DOI: https://doi.org/10.7554/eLife.43510.004

## Within-host and population diversity are spatially variable and reflect transmission intensity

Within-host diversity and population level genetic diversity were assessed by health district (n = 4) and health facility catchment (n = 29). Across health districts, infections from Rundu and Andara had greater within-host diversity than those from Nyangana and Zambezi districts as demonstrated by higher multiplicity of infection (MOI) and lower within-host fixation index (*Figure 2—figure supplement 1*), consistent with the higher malaria incidence and higher proportion of imported malaria cases in these districts. Overall, the genetic diversity of the parasite population was high throughout the study area (median $H_E$ = 0.79 [IQR: 0.60–0.85]), though lower in Zambezi than the other three health districts (*Figure 2—figure supplement 1*). When stratified by health facilities, the patterns of within host and population diversity showed variability within districts, providing supporting evidence of fine-scale heterogeneity of malaria transmission in the study area (*Figure 2—figure supplement 2*). For example, infections detected at Rundu district hospital had the highest within-host diversity, which may be due to the larger proportion of patients who traveled to or resided in Angola (reported by 13% of patients). Infections from Rundu also had the highest population diversity, which may be attributable to the large catchment area of this facility (48% came from beyond the study region). Cases were then classified as local or imported based on recent travel history and the location of residence. Infections from individuals with a history consistent with importation had higher within-host diversity than those without, despite having similar population-level diversity (*Figure 2A–C*). These data suggest a lower rate of superinfection and thus less local transmission in northeastern Namibia compared to the international source populations.

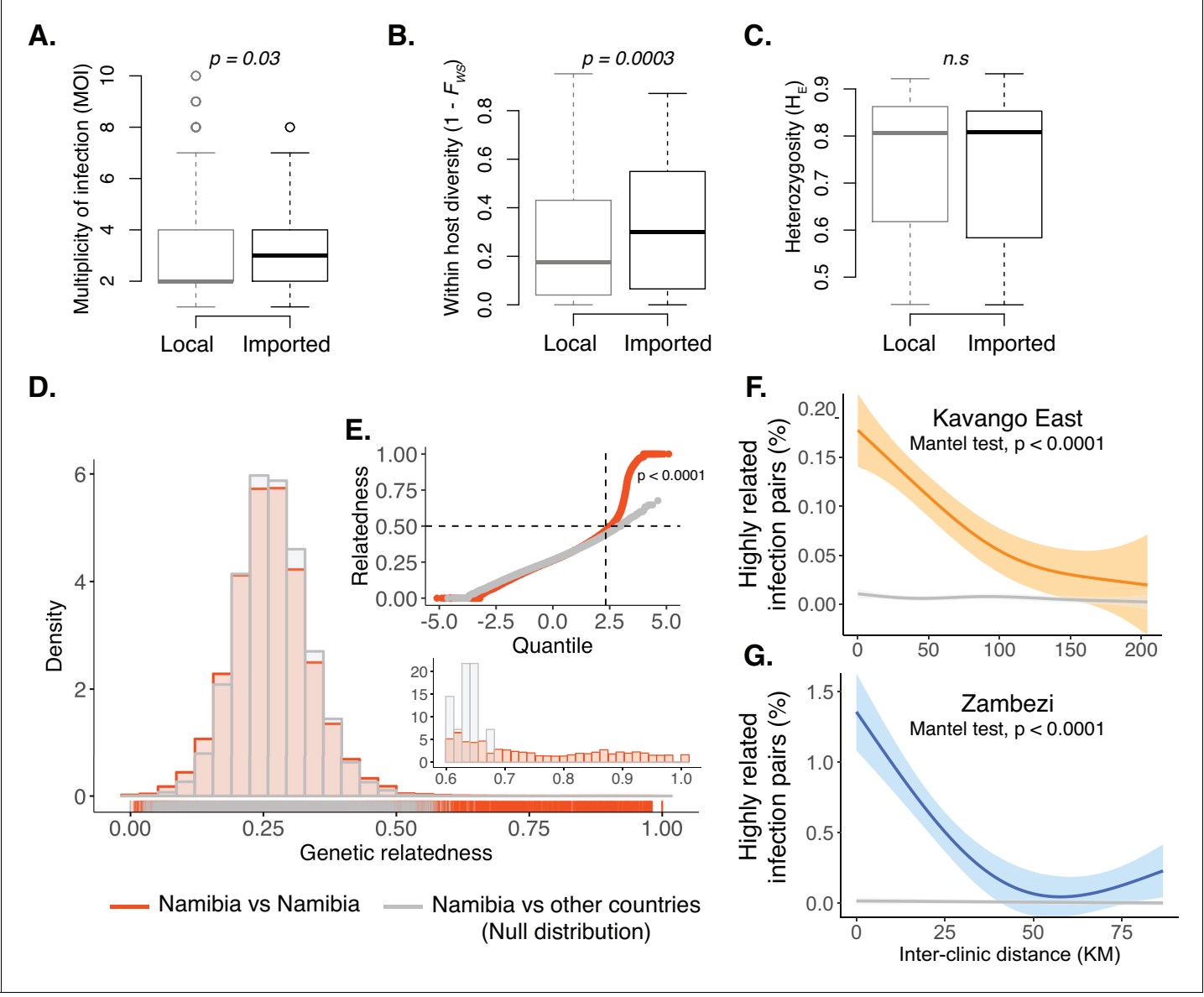

**Figure 2.** Within-host diversity, heterozygosity and genetic relatedness. (A) Multiplicity of infection (MOI); (B) within-host diversity index (1-$F_{WS}$) and (C) population level genetic diversity (heterozygosity, $H_E$) compared between potentially imported samples (black) and those without any evidence of being imported (gray). $F_{WS}$ is analogous to an inbreeding coefficient. A 1-$F_{WS}$ value shows outbreeding and a value of 0 indicates a single clone infection. Population level genetic diversity was measured as the distribution of heterozygosity in 26 microsatellites. (D) Pairwise genetic relatedness between samples was calculated using the identity by state (IBS) metric including all alleles detected in polyclonal samples. Highly related infection pairs were identified based on a null distribution, pairwise relatedness between samples from Namibia and other countries from West, Central and East Africa. The inset shows a zoomed in histogram of pairwise genetic relatedness between samples with genetic relatedness ≥0.6. (E) The quantile plot indicates the divergence of the distributions at genetic relatedness ≥0.5 (n = 20,988 pair-wise comparisons between infections collected from Namibia). The x-axis is the z-score values of the population quantiles of the distributions. The dashed vertical line corresponds to the 99% percentile of the distribution. (F and G) The relationship between highly related infections (i.e. number of pairs with a genetic relatedness ≥0.6/ total number of pairs) and the inter-clinic distance in Kavango (F) and Zambezi (G). Geographically adjacent infections were more highly related than those at further distances. The shaded areas show the 95% confidence interval. The gray line indicates a null distribution created by bootstrapping (n = 1000) over the inter-clinic distance.

DOI: https://doi.org/10.7554/eLife.43510.005

The following figure supplements are available for figure 2:

**Figure supplement 1.** Within-host and population diversity by health district.

DOI: https://doi.org/10.7554/eLife.43510.006

**Figure supplement 2.** Within-host and population diversity by health facilities.

*Figure 2 continued on next page*

*Figure 2 continued*

DOI: https://doi.org/10.7554/eLife.43510.007

**Figure supplement 3.** Existing methods reveal no parasite population structure in northern Namibia.

DOI: https://doi.org/10.7554/eLife.43510.008

**Figure supplement 4.** Relationship between pairwise genetic differentiation and inter-clinic distance between health facilities in two regions of northern Namibia.

DOI: https://doi.org/10.7554/eLife.43510.009

## Population structure and differentiation within northeastern Namibia

Existing model and distance (multidimensional scaling and phylogenetic tree) based methods did not identify any spatial clustering between health facilities or health districts (*Figure 2—figure supplement 3*). Population measures of genetic differentiation ($G_{ST}$ and *Jost's D*) also showed no relationship with geographic distance between health facilities (*Figure 2—figure supplement 4*). However, a novel analysis evaluating the distribution of genetic relatedness between all infections, including polyclonal infections, revealed a strong spatial signal. For this analysis, we identified highly related pairwise connections (identity by state, IBS) between health facilities, using comparisons between Namibia and non-neighboring African countries as a null distribution. The distribution of the pairwise genetic relatedness within the study area diverged from the null distribution at $\geq 0.5$, and these highly related infection pairs were responsible for the majority of the spatially informative genetic signal (*Figure 2D and E*). We found a decay in genetic relatedness with increasing geographic distance within each of the two regions of Namibia (*Figure 2F and G*, p<0.0001, Mantel test), suggesting that there was sufficient sustained local transmission occurring in both study areas to create a strong spatial gradient in parasite populations.

## Local transmission and genetic connectivity within northeastern Namibia

To evaluate the local connectivity of parasite populations, we computed the pairwise genetic connectivity between infections sampled from different health facilities. In Zambezi, 60% (9/15) of the pairwise connections were highly related. However, we observed overall lower connectivity and fewer highly related pairwise connections 39% (99/253) in the Kavango East region (*Figure 3A*, *Figure 3—source data 1*). The degree of parasite connectivity between health facility catchments was heterogeneous, with some very well-connected health facilities (i.e. highly related connections to most other facilities) and others relatively unconnected. For example, two health facilities in Zambezi and four in Kavango East were connected to most other health facilities in each region (connectivity score = 0.65–0.91, *Figure 3B*, *Figure 3—source data 1*). In Zambezi, the two most connected clinics also had the highest incidence of malaria (*Mumbengegwi et al., 2018*) and were in close proximity to the Angolan border and Kavango East region than the other clinics. In contrast, Rundu district hospital, the largest health facility in the study area, was only connected to a few other health facilities (connectivity score = 0.05), consistent with the high genetic diversity and large catchment of this hospital, extending well beyond the study area. Overall, 17% of the pairwise genetic connectivity measures between health facilities in Kavango East and Zambezi regions were highly related, providing evidence of mixing between these parasite populations. Health facilities that were the most genetically connected within a region were also the most connected between regions (*Figure 3C*), suggesting that specific localities may represent priority targets for interventions to efficiently reduce within and between region transmission.

## Estimated connectivity from human mobility data

We also sought to estimate parasite connectivity using human mobility data. Estimates of time at risk for infection with malaria parasites were quantified from travel surveys and population estimates of human movement within Namibia derived from mobile phone calling data. Few individuals reported at least one night spent away from their residence location (mean 1%), with a higher percentage (17%) of mobile phone subscribers' overall time spent outside of their primary residence tower (*Figure 1C*). To estimate parasite mixing patterns from both data sets, we calculated the proportion of time spent at all destinations scaled by the relative malaria incidence to create a

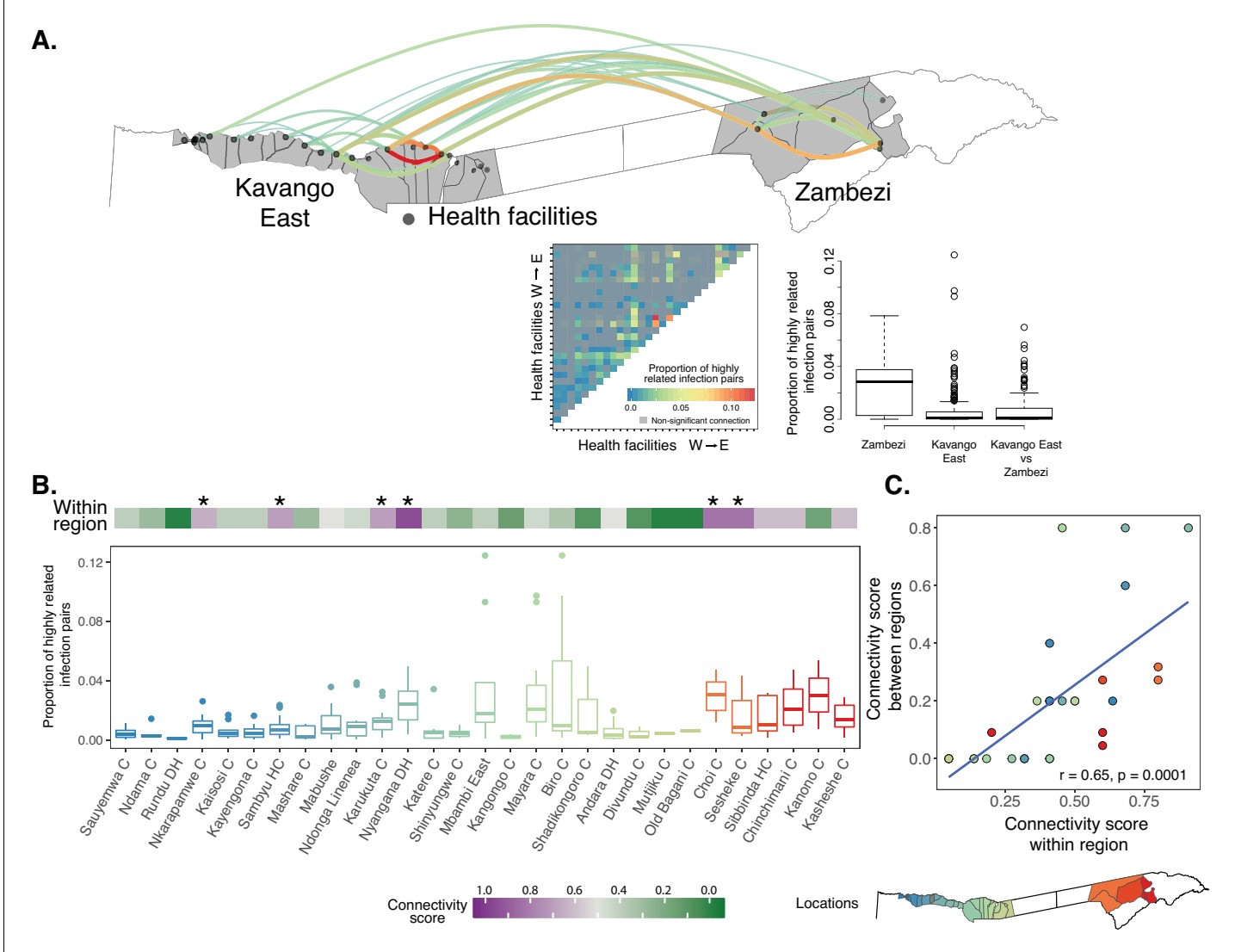

**Figure 3.** Local genetic connectivity in northeastern Namibia. (**A**) Proportion of highly related infections between 29 health facilities in Kavango East and Zambezi regions is shown. The heat map shows 406 pairwise proportions of highly related infections between health facilities. Highly related values are shown in color, all other values are shown in gray. Top 10% of the pairwise connections are shown on the map of the study area. Highly related connections were determined after correcting for multiple comparisons using a Bonferroni correction. The boxplot illustrates that infections within Zambezi were more related to each other than to Kavango East. (**B**) Health facilities in Kavango East and Zambezi are shown from west to east. The connectivity score (top heatmap) ranges from 0 to 1, where a score of 1 indicates a health facility is highly connected to all other health facilities in the region. The proportions of highly related infections among the health facilities is shown to illustrate the strength of these connections (boxplot). Health facilities with the highest connectivity are indicated by a star (Nkarapamwe, Sambyu, Nyangana, and Karukuta health facilities in Kavango East and Choi and Seseheke health facilities in Zambezi). Scatterplot shows comparison of within region and between region connectivity. Boxplots and points are colored to indicate the geographic location of the health facility catchment on the inset map.

DOI: https://doi.org/10.7554/eLife.43510.010

The following source data is available for figure 3:

**Source data 1.** Proportion of highly related infections and connectivity scores by health facility.

DOI: https://doi.org/10.7554/eLife.43510.011

proportion of time at risk measure for each individual that was then aggregated to quantify mixing between locations. In both travel survey and mobile phone data, individuals spent the majority of their time at their location of residence, and mixing patterns inferred from both data sources found

that individuals spent similar amounts of time at risk for importing malaria to Kavango East and Zambezi regions (*Figure 4—figure supplement 1*).

## Mixing of parasite populations inferred from parasite genetic and human mobility data

To compare the various data sets at equivalent spatial scales, we aggregated the genetic data to mobile phone catchments (n = 14) and travel survey destination locations (n = 8). In the mobility data alone, both mobile phone catchments (n = 14) and travel survey destination locations (n = 8) were strongly connected to their neighbors, with few travelers between Kavango East and Zambezi (*Figure 4*). Clusters identified using modularity maximization of these networks also highlight a strong spatial signature, where contiguous locations formed clusters. Using the same procedure and spatial areas, clusters were identified for the aggregated parasite genetic data. Clusters identified from mobile phone data shared some similarity to those identified from the genetic data (grouping similarity measure: Rand Index = 0.76, *Figure 4* and *Figure 4—figure supplement 2*, *Figure 4—source data 1*). However, genetic data identified a substantial amount of parasite connectivity between locations that was not detected by mobile phone data. There was little agreement between clusters identified from the travel survey data and genetic data (grouping similarity measure: Rand Index = 0.46), albeit limited by a smaller number of geographic units due to the coarser spatial scale of the travel data. The precision of results obtained from travel data was also limited by the relatively small number of individuals who reported travel within the study area. Overall, these results suggest that both travel survey and mobile phone data have limitations in capturing the structure of parasite connectivity within northeastern Namibia detected using genetic data.

## Evidence of cross-border connectivity between Namibia, Angola and Zambia

To evaluate cross-border connectivity, geographic regions were aggregated to nine locations: four health districts in Namibia, three locations in Angola, and two provinces in Zambia. Mobile phone data were limited to Namibia, not allowing for evaluation of cross-border connectivity. Data from the travel survey identified some sources of cross-border importation into Namibia, with the most prominent connections being from Rundu to southern Angola (2.8% of cases reporting travel to this area) and Zambezi to Western Zambia (1.8% of cases, *Figure 5A*). However, these data were only able to identify symptomatic cases with a direct history of travel and would not identify any cases which failed to report relevant history or those cases which may have originated from importation via asymptomatic carriers or transmissible individuals otherwise not detected by the routine surveillance. Therefore, we augmented travel history with genetic data to estimate the underlying connectivity of the parasite populations from the same nine locations.

To evaluate connectivity using parasite genetics, we analyzed genetic data collected from Namibia (this study) as well as additional data from Angola and Zambia. Infections with malaria parasites from Namibia and northern Angola were not closely related (mean proportion of highly related infections = 0.00004 [Range = 0–0.00015]). However, parasites between health districts of Namibia; between Namibia and southern Angola; and between Namibia and Western and Southern provinces of Zambia were more closely related (*Figure 5B*). Overall, infections from Namibia were on average 142 times more likely to be genetically related to those from southern Angola and 191 times to Zambia than to those from northern Angola, indicating substantial parasite mixing within the geographically connected Namibia-Angola-Zambia regional block. In contrast, this finding suggests limited parasite connectivity between northern and southern Angola, though limited, non-contemporaneous sampling within Angola makes it difficult to make more detailed conclusions about transmission in this country. Within Namibia, infections from Andara and Nyangana were 3 and 4 times more likely to be genetically related to Zambezi than infections sampled from nearby Rundu, respectively (*Figure 5B*).

When evaluating specific connections between the nine locations, we estimated the direction of parasite flow in addition to the degree of connectivity between areas by weighting the pairwise proportion of highly related infections by malaria incidence to account for the differential risk of infections in different areas. Results from this analysis provided substantially more information on regional connectivity than using travel history data alone. The four studied health districts within

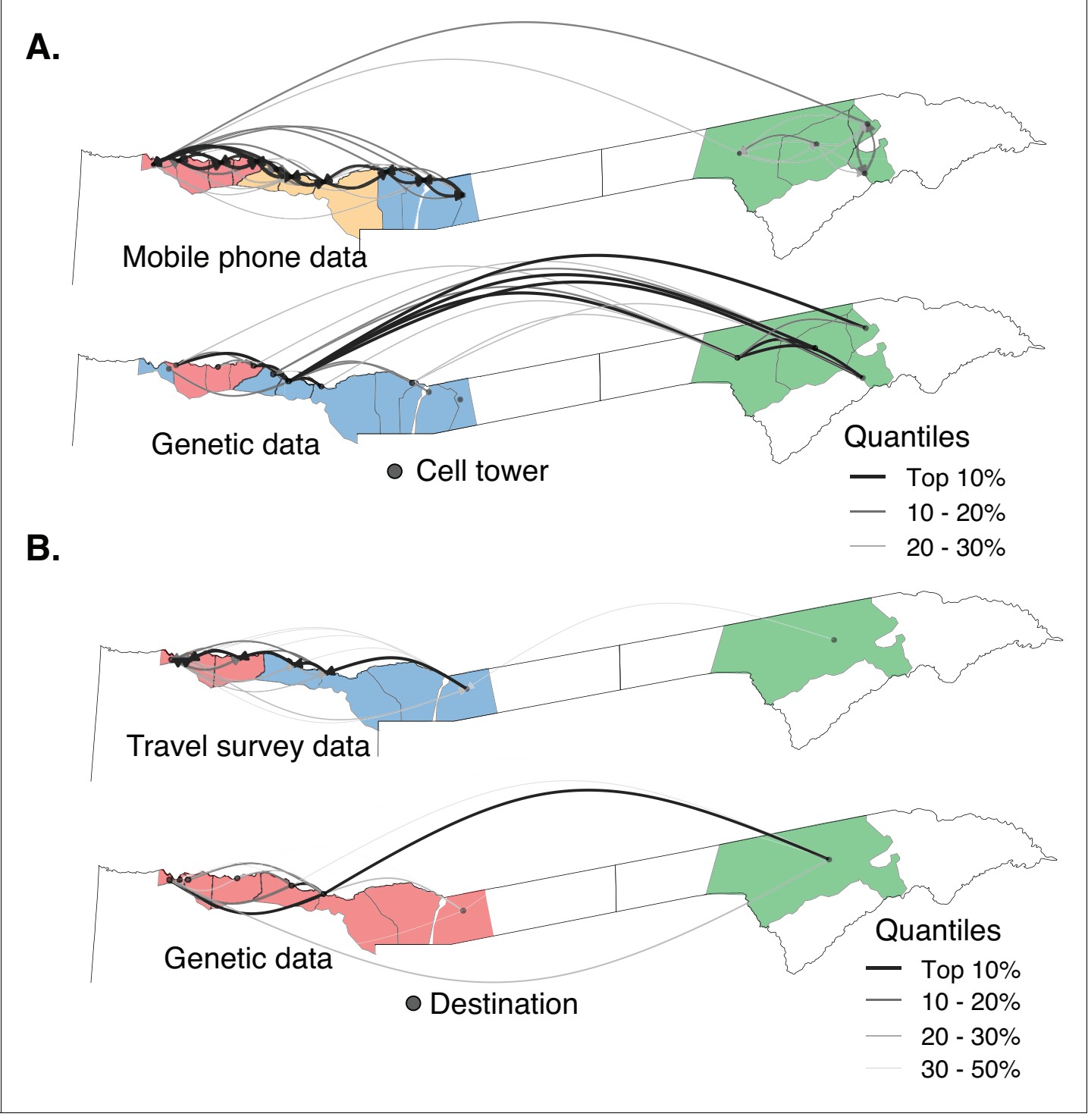

**Figure 4.** The relationship between parasite connectivity estimated from the two sources of mobility data and parasite genetic data. Mobility-based (top row) and parasite genetic (bottom row) clusters were identified using mobility estimated from the (**A**) mobile phone data and (**B**) travel survey data and genetic data aggregated to the level of the respective mobility data. The top routes of mixing by human mobility and connectivity by parasites genetic data are shown. For the genetic data, highly related connections are shown. Catchment areas are colored with identified clusters from each data type (see *Figure 4—figure supplement 2*). Genetic data identified a greater number of long distance connections than either estimate from human mobility data.

DOI: https://doi.org/10.7554/eLife.43510.012

The following source data and figure supplements are available for figure 4:

*Figure 4 continued on next page*

*Figure 4 continued*

**Source data 1.** The relationship between parasite connectivity estimated from the two sources of mobility data (i.e. mobile phone and travel history) and parasite genetic data.
DOI: https://doi.org/10.7554/eLife.43510.015
**Figure supplement 1.** Human mobility and parasite mixing based on travel survey and mobile phone data in two regions of northern Namibia.
DOI: https://doi.org/10.7554/eLife.43510.013
**Figure supplement 2.** The relationship between parasite connectivity estimated from human mobility and parasites genetic data sources.
DOI: https://doi.org/10.7554/eLife.43510.014

Namibia were connected to each other but had stronger connections to nearby cross-border locations than to each other (*Figure 5C and D*, *Figure 5—source data 1*). The most important source populations were Calai and Dirico (border towns in southern Angola), followed by Western and Southern provinces of Zambia. Although there was evidence of importation into both Kavango East and Zambezi regions from other countries, based on these data the Zambezi region was estimated to receive high rates of importation from a larger number of sources (i.e., it was a dominant sink population).

## Discussion

It is clear that achieving elimination of malaria will require strategic coordination of local and regional interventions guided by accurate intelligence on parasite movement; what has not been clear is how to best obtain this information. In this study, we demonstrated that augmentation of traditional malaria surveillance with parasite genetic data added substantially to the understanding of transmission epidemiology in a critical region of Southern Africa straddling the border between Namibia, Angola, and Zambia. First, parasite genetics provided positive evidence that the majority of malaria cases observed in northeastern Namibia were due to local transmission, evidenced by the strong fine-scale spatial structure in the genetic data. It would be difficult to explain such consistent spatial clustering of highly related parasites and the observed decay with distance if local transmission did not predominate. This is a key piece of programmatically relevant information that would have been difficult to confirm with negative evidence, that is merely based on a lack of history consistent with importation, especially given potential disincentives for individuals to report living outside of Namibia. Second, the addition of genetic data to travel histories provided more detailed and extensive evidence of parasite connectivity over hundreds of kilometers, both within Namibia and across borders from Angola and Zambia. Conclusions regarding the origins and relative magnitude of malaria importation from genetic data were distinct from those obtained from travel history alone, which were likely limited by sparsity and bias, and from mobility from the mobile phone data, which were unable to inform cross-border movement in this study and appeared to underestimate the importance of long-distance connections within Namibia. Although cross-border movement is possible to obtain from mobile phone calling data, for example if the handset ID was used instead as an anonymized ID, data available for this study were limited to national travel patterns.

Malaria programs will require targeted interventions at sub-national scales to effectively achieve and sustain elimination (*WHO, 2017*). The success of such programs, for example targeted vector control, focal screening and treatment, and mass drug administration will largely be dependent on tailoring interventions to drivers of ongoing transmission (*WHO, 2014*). Using a novel analytic framework, we found that despite a signal of predominantly local transmission (within tens of kilometers), parasite populations in Namibia remain highly connected at longer scales within and between the two administrative regions (over hundreds of kilometers). In this context, restricting interventions to a relatively small area, such as a region, may result in improved malaria control but is unlikely to achieve elimination unless any malaria transmission from imported parasites is completely prevented. Indeed, limiting elimination efforts to national boundaries may be doomed to fail for the same reasons, for example it may be necessary for Namibia to coordinate efforts with Angola and Zambia to eliminate transmission within its own boundaries (*Khadka et al., 2018*). Consistent with this hypothesis, we found that parasite populations within Namibia were in many cases more closely connected to those across the border than to other parasites from the same country. At a more nuanced level, variations in the degree of genetic connectivity between areas we observed could be used to

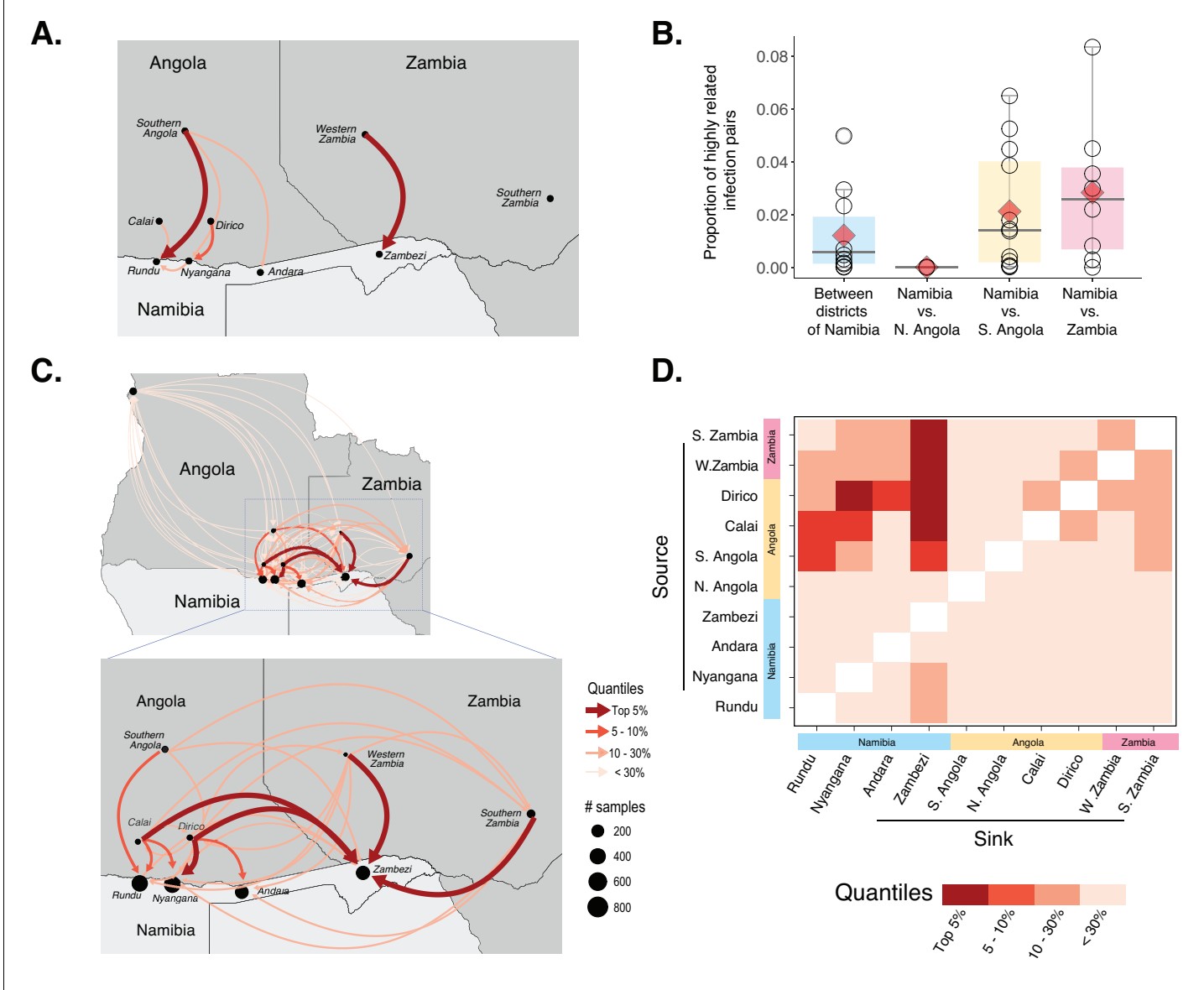

**Figure 5.** Cross-border connectivity estimated from genetic and travel survey data. (A) Cross-border connectivity estimated from travel survey data. Importation was estimated from the proportion of individuals who reported travel to each destination weighted by the ratio of malaria incidence from local health system data at a destination to the residence location (see Materials and methods). The importation estimate was visualized on the map and colored by quantiles (lowest values - light pink, highest values - dark red). The arrows indicate the direction of malaria importation. Three individuals reported travel beyond the scope of the map: two individuals, from Andara and Zambezi, reported travel to northern Angola and one individual from Zambezi reported travel to Northern Province of Zambia. (B) Samples across Namibia, Angola, and Zambia were genotyped and the proportion of highly related infections are shown between health districts of Namibia and between Namibia and northern Angola, southern Angola and Zambia. The mean proportions of highly related infections are indicated by red diamonds. (C) Importation estimates and directionality of parasite flow was estimated from genetic data along with malaria incidence values in the pairs of locations indicated. Estimates of importations between locations are shown by quantiles (lowest values - light pink, highest values - dark red). The arrows indicate the direction of malaria importation. Estimates of importations were based on the proportion of highly related infections between pairs of location weighed by the ratio of malaria incidence between the two locations (See Materials and methods: *Estimates of cross-border importation and connectivity).* Locations in Angola are indicated for centroid location of Calai area, Dirico area and the Cuando Cubango province (i.e. southern Angola). For Namibia, centroid locations of the health districts were indicated. For Zambia, centroid locations of Western and Southern provinces are shown. Northern Angola was not strongly related to samples in the study area and corresponding cross-border region, suggesting that parasite flow between southern and northern Angola is less than that between (more geographically proximate) cross-border regions of northern Namibia and southern Angola. (D) Heat map showing sources and sinks in the Namibia-Angola-Zambia block. Within the regional block, locations within Namibia were related to locations across the border, with higher estimated

*Figure 5 continued on next page*

*Figure 5 continued*

parasite flow to Zambezi from these locations than to Kavango East. Interestingly, locations in Zambezi demonstrated more connectivity to Calai, Dirico, Western and Southern Zambia than locations in Kavango East.

DOI: https://doi.org/10.7554/eLife.43510.016

The following source data is available for figure 5:

**Source data 1.** Importation estimates from genetic data.

DOI: https://doi.org/10.7554/eLife.43510.017

optimize the order in which interventions may be most efficiently implemented. For example, all else being equal, targeting interventions to locations with greater genetic connectivity than those with lower connectivity may be more effective in fragmenting the parasite population and reducing the influx of malaria from pockets of transmission. This finding is consistent with the previous observation that malaria cases were clustered in northeastern Namibia (*Smith et al., 2017*; *Tatem et al., 2014*). In addition, enhancing malaria surveillance and access to care, for example through additional clinics or border posts, may be effective in reducing the extent of cross-border importation if deployed in areas with high measured importation rates.

We estimated malaria parasite connectivity from mobile phone data, travel history, and parasite genetics, allowing us to compare estimates based on human population movement to more direct estimates of the connectivity of parasite populations. In principle, anonymized mobile phone data can provide a continuous and inexpensive source of human mobility data. Within Namibia, connectivity estimates derived from vast amounts of mobile phone data roughly mirrored those derived from genetics, though they were predominated by small scale movements and did not capture the extent of longer distance connections revealed by the genetic data. This difference could be due to differential patterns of movement of people with access to mobile phone and those who potentially transmit malaria, limitations in the accuracy of malaria incidence estimates used as inputs in the model, or the temporal difference between the two data sets (*Ruktanonchai et al., 2016*). An important current limitation of call data records analyzed was the inability to provide any information on movement beyond national boundaries.

In contrast to mobile phone data, travel data collected from symptomatic cases were sparser, estimated less travel, and were collected at a coarser spatial scale, limiting agreement with the genetic data within Namibia. However, travel data were able to provide low-resolution information and demonstrated evidence of cross-border importation by infected individuals, albeit possibly biased by patient omission on international travel or residence. Estimates derived from genetics are likely to be more comprehensive, as even with perfect accuracy travel history only captures movements of the interviewed patient while genetics can record evidence of movement through multiple generations of transmission. Importantly, information on travel allowed us to greatly extend the utility of genetic data, providing a means of 'sampling' parasites from beyond the study site in those with a definitive history of international travel. Travel history remains a critical part of routine surveillance, and when collected reliably and ideally at finer spatial scale than done here will likely provide important information on its own (*Smith et al., 2017*; *Tejedor-Garavito et al., 2017*) and in conjunction with genetic data.

The genetic data used in this study were generated via traditional methods – a panel of 26 microsatellites – but were well-suited for the intended application and captured strong spatial signal over local and regional scales. Particular strengths of these data were the ability to capture information from polyclonal infections (77% of the study population) given the multiallelic nature of the loci, and to obtain robust results from easily collected field samples (dried blood spots and used rapid diagnostic tests). Targeted deep sequencing of short, multiallelic haplotypes may provide similarly rich data from polyclonal infections, allowing greater flexibility in the number and location of loci, facilitating easier comparisons across data sets, and taking advantage of continual advances and cost savings in sequencing technology (*Aydemir et al., 2018*; *Lerch et al., 2017*). Generating *P. falciparum* whole genome sequence data from Southern Africa would facilitate rational selection of the most informative sequence targets for local and regional parasite movement.

Advances in analytical methods would likewise improve the quality of information obtained from parasite genetics (*Wesolowski et al., 2018*). Our methods for computing genetic relatedness, like

the genetic data themselves, were relatively simple but provided useful information on relative connectivity between geographic areas. In this study site, classical methods for measuring or visualizing genetic differentiation (e.g., Gst, STRUCTURE, or phylogenetic trees) had limited utility due to the marginal differences in allele frequencies between geographically proximal locations in this relatively compact study site and the inability to utilize all information from polyclonal infections. Currently, there are few established analytical approaches to quantify fine-scale genetic connectivity between locations with predominantly polyclonal infections. Estimation of pairwise genetic relatedness between all infections in this study allowed the incorporation of data from all parasites detected in infections and extraction of useful signals of recent transmission created by recombination and cotransmission of multiple parasites. However, more sophisticated bioinformatics, statistical, and modeling tools that are designed to take advantage of genetic data from polyclonal infections and map these onto quantitative, calibrated estimates of migration rates would transform the utility of genetic data for understanding operationally relevant transmission patterns. The availability of such tools would provide a strong rationale for coordinated collection of regional data on parasite genetics, allowing for more systematic evaluation of malaria transmission and generalized utility.

The incorporation of parasite genetics added an important dimension to the understanding of local and cross-border malaria transmission epidemiology and connectivity in this area of the Elimination 8 region of Africa. Our results, showing strong connectivity between malaria parasite populations over hundreds of kilometers within Namibia and across national borders, calls for strengthening the simultaneous coordination of efforts between the Elimination eight countries. Furthermore, our data demonstrate the feasibility and added value of systematically integrating genetic data into national and regional surveillance efforts, particularly when the goal is elimination and movement of malaria parasites may threaten this goal or influence interventions. A combination of human mobility and parasite genetic data is proposed to mitigate limitations of each individual data source in isolation and to provide the most robust intelligence to guide local and regional strategy.

## Materials and methods

### Ethics statement

Ethical approval for the study was obtained from the Institutional Review Boards of the University of Namibia and the University of California, San Francisco (Identification numbers 15–17422 and 14–14576). Informed consent was obtained from all participants or the parents of all children participated in the Zambezi study. For the Kavango study, IRB approval was obtained but no informed consent was collected as all samples (used RDTs) and de-identified data were collected during routine surveillance.

### Study design and participants

We enrolled 4643 symptomatic *Plasmodium falciparum* cases from the outpatient clinics of 29 health facilities in two regions of northeastern Namibia: Kavango East and Zambezi. Diagnosis of all cases was confirmed by rapid diagnostic test (RDT). In Kavango East, 3871 symptomatic cases from 23 health facilities were enrolled and used RDTs were collected from March to June 2016. In the Zambezi region, 772 symptomatic cases from six health facilities were enrolled between February 2015 and June 2016 and dried blood spots (DBS) were collected at the time of diagnosis. In both locations, additional patient information such as age, residence, local and international travel history were collected. The health facilities in Kavango East and Zambezi were located within 204 km and 87 km of each other, respectively.

### Human mobility data

During the time of RDT or DBS collection, study participants were asked about their location of residence as well as any overnight travel to non-residence locations. In the Kavango travel survey, individuals were asked if there was any travel to a select number of locations including neighboring towns, other districts/provinces in Namibia and Angola, and other neighboring countries. In the Zambezi travel survey, individuals were able to provide information about travel to any location. These free-response questions were geocoded to the village and regional levels and included both national and international destinations. Individuals were also asked to provide information on the

duration of the trip (in days). When analyzing travel survey data, each individual's time over the prior 30 days was allocated based on their location of residence and the reported time spent away from their residence. To compare with the genetic data, health facility catchments were aggregated to the corresponding travel survey location based on the location of the catchment centroid.

Mobile phone call data records were obtained from October 2010 to September 2011. In total, 1.19 million unique individual subscribers were recorded at 197 mobile phone towers in Namibia (*Ruktanonchai et al., 2016*). Of these, 14 towers with a total of 98,104 subscribers were located within the study area. Travel patterns between mobile phone tower catchment areas were calculated using previously developed methods (*Ruktanonchai et al., 2016*; *Tatem, 2014*; *Wesolowski et al., 2012*). Briefly, individuals were assigned a primary mobile phone tower based on the most frequently used tower at night. Trips to other mobile phone tower catchments were inferred if their primary daily location was recorded at another tower and was not limited to only night time use. All other time was assumed to be spent at their primary mobile phone tower. Individuals were aggregated to a single primary tower location, and mobility per mobile phone tower catchment was calculated as a distribution of time spent at each one of the other mobile phone tower catchments, including the time spent at the primary tower location. Mobile phone tower catchments and health facility catchments, although covering the same geographic area, did not correspond to a one-to-one match. When comparing mobility data from the mobile phone tower catchments with the genetic data, health facilities were aggregated to tower catchments based on the location of the catchment centroid.

## DNA extraction and genotyping

DBS and used RDTs were stored with desiccant at −20°C until transportation and processing. DNA was extracted from 6 mm punches of DBS and strips of used RDTs using the Saponin-Chelex method (*Plowe et al., 1995*). For RDTs, the cassettes were opened using a thin metal spatula and DNA was extracted from the nitrocellulose strip in accordance with the worldwide antimalarial resistance network guidelines (*Molecular Module, 2011*), with the exception that DNA extraction was performed in deep 96-well plates. For all samples extracted from DBS, parasite density was quantified using *var*-ATS ultra-sensitive qPCR (*Hofmann et al., 2015*) and samples with more than 10 parasites/μL of blood were genotyped. Given the large number of RDT samples collected, a subset was selected for extraction and genotyping as follows. If less than 100 RDTs were collected from a given clinic, all were genotyped. If more than 100 RDTs were collected from a given clinic, any cases with travel history and 100 cases without travel history were genotyped. In addition, all samples were genotyped from one hospital (Nyangana Hospital) to validate subsampling. For samples extracted from RDTs, parasite density was quantified on a subset. When positive, parasite density was almost always above the genotyping threshold (n = 320, median = 13612 parasites/μL of blood). A total of 2990 samples were genotyped using 26 microsatellite markers as described previously (Liu et al., under preparation). Briefly, two-rounds of PCR protocol were used to amplify the 26 microsatellite loci. The multiplex primary PCR was performed in 4 groups using two different PCR conditions. 1 μL of the amplified product was then used as a template for the individual PCR for each marker. PCR products were then diluted and sized by denaturing capillary electrophoresis on an ABI 3730XL analyzer with GeneScan 400HD ROX size standard (Thermo Fisher Scientific). The resulting electropherograms were analyzed using microSPAT software (*Murphy, 2018*) to automate identification of true alleles and differentiate real peaks from artifacts. A total of 2585 samples with data in at least 15 or more loci were included in these analyses (S1 Data). Additional data from comparably genotyped microsatellite datasets from northern Angola (from Cabinda, Bengo, Uige and Zaire provinces) collected between January and December 2014 (n = 137, Liu et al., under preparation) and southern Zambia (from Choma district) collected between January 2015 and April 2016 (n = 96, *Pringle et al., 2018*) were also analyzed. Genotyping data from all samples were combined and processed with similar software settings to avoid variability in allele calling.

## Within-host and population level genetic diversity

The within-host diversity of infections was determined using multiplicity of infection (MOI) and the $F_{WS}$ metric. MOI was determined as the second highest number of alleles detected at any of the 26 loci, allowing for the possibility of false positive allele calls. The $F_{WS}$ metric is a measure of the

within-host diversity of an individual infection relative to the population level genetic diversity. A high $F_{WS}$ indicates low within-host diversity relative to the population (e.g. low risk of inbreeding). $F_{WS}$ was calculated as described previously (*Roh et al., 2019*; *Auburn et al., 2012*). Briefly, $F_{WS}$ was calculated for each infection using the formula, $F_{WS} = 1 - \frac{H_w}{H_s}$ where $H_w$= heterozygosity of the individual and $H_s$= heterozygosity of the local parasite population. Within host heterozygosity was estimated based on the number of alleles detected at each locus. Mean $F_{WS}$ was calculated for each individual by taking the mean across all loci. Population level genetic diversity was estimated using expected heterozygosity ($H_E$) and calculated using the formula, $H_E = \left[\frac{n}{n-1}\right]\left[1 - \sum p_i^2\right]$, where n is the number of genotyped samples and $p_i$ is the frequency of the $i^{th}$ allele in the population. Within-host and population level genetic diversity were then compared by health districts, health facilities and between local and imported cases. Imported cases include residents of Angola and Zambia and those individuals with a reported travel history to Angola and Zambia in the last 30 days.

## Pairwise genetic relatedness between infections

Methods for computing genetic relatedness between infections, incorporating data from all alleles detected at a loci, are lacking due to the difficulty of accurate reconstruction of haplotypes from polyclonal infections. Most existing methods either rely only on haplotypes constructed in monoclonal infections or 'reconstructing' haplotypes from only the dominant alleles in polyclonal infections (i.e. not utilizing all the alleles detected in a polyclonal infection). In this study, we computed allele sharing between pairs of infections, allowing us to utilize all the detected alleles at a loci in polyclonal infections. For all successfully genotyped samples, pairwise genetic relatedness between infections was calculated using a modified identity by state (IBS) metric (*Jacquard et al., 1974*; *Pringle et al., 2018*). Briefly, IBS was computed based on the number of shared alleles between pairs of infections, in both mono- and poly-clonal infections. The overall pairwise IBS was calculated as:

$$IBS = \frac{1}{n}\sum_{i=1}^{n}\frac{S_i}{X_i Y_i}$$

where $n$ is the number of genotyped loci, $S_i$ is the total number of shared alleles at locus $i$ between samples $X$ and $Y$; $X_i$ is the number of alleles in sample $X$ at locus $i$ and $Y_i$ is the number of alleles in sample $Y$ at locus $i$. Within the Namibia dataset, a total of 3,365,700 pairs of infections from 29 health facilities were analyzed.

## Population structure and genetic differentiation

To investigate geographic clustering, individuals were aggregated to health districts. Population structure was inferred to determine whether haplotypes (estimated from dominant alleles) clustered into distinct genetic populations (K) using the software MavericK (*Verity and Nichols, 2016*). Clustering was further evaluated by a neighbor-joining phylogenetic tree computed using the 'ape' package (*Paradis et al., 2004*) and PCA analysis using the pairwise genetic distances (1-IBS) determined above. *Jost's D* (*Jost, 2008*) and $G_{ST}$ (*Nei and Chesser, 1983*) were used to estimate genetic differentiation between pairwise comparisons of clinics. Briefly, *Jost's D* and $G_{ST}$ were calculated using the formulas: $D = \left[\frac{H_T - H_S}{1 - H_S}\right]\left[\frac{n}{n-1}\right]$ and $G_{ST} = (H_T - H_S)/H_T$, respectively, where $H_T$ and $H_S$ are the overall and the sub-population heterozygosity, respectively and n is the number of sampled populations. The values of *Jost's D* and $G_{ST}$ range from 0 (no genetic differentiation between populations) to 1 (complete differentiation between populations).

## Determining highly related infection pairs and connectivity

To investigate connectivity at different spatial scales, highly related infection pairs were identified. In order to determine pairs of infections which were more related than expected by chance, we used genotyping data with a similar MOI distribution from countries in West, Central and East Africa (n = 432, data from Liu et al., under preparation). These countries are not geographically connected to the study area, thus there is likely limited direct parasite connectivity. The distribution of pairwise genetic relatedness between these and Namibia samples was estimated and used as the expected distribution of relatedness in the absence of a direct transmission link and/or a recent importation

event (i.e., a null distribution). For each pair of locations and the null distribution, IBS values were binned into 20 bins. For each bin, the difference in the proportion of observed and expected pairs under the null distribution was computed. The last bin at which the observed proportion was greater than the null distribution, starting from 1 to 0, was used as a cut-off to determine the proportion of highly related infections. To investigate spatial connectivity between locations, the proportions of highly related infections above the cut-off were compared. The median of the cut-off was an IBS of 0.55. The overall proportion of highly related infections was calculated as the sum of the proportions of observed pairs above the estimated cut-off minus the proportion in the null distribution. The statistical significance of connectivity was determined by bootstrapping over the IBS values 1000 times and correcting for multiple comparisons using a Bonferroni correction, generating a 95% confidence interval for each pair of locations.

## Estimates of within-country importation and connectivity

In total, there were eight travel survey destinations and 14 mobile phone towers that overlapped with the health facilities of the study area. We scaled the time spent, estimated from either data set, based on the ratio of incidence in the destination versus the corresponding health facility (*Figure 4— source data 1*). When multiple health facilities fell within a single travel survey destination or mobile phone catchment, the average incidence was used. These data were used to estimate the proportion of time at risk and possible source locations of importations for each health facility (measure of parasite mixing). Clusters were determined using a hierarchical modularity maximization algorithm (*Newman, 2006*) from either the incidence scaled travel between travel survey destinations, mobile phone tower catchments, or the proportion of highly related samples from the genetic data. We clustered the genetic data from Kavango and Zambezi separately in order to identify sub-regional structure in Kavango. For both the travel or parasite mixing data, Kavango and Zambezi were able to cluster together. We then compared the cluster agreement estimated from mixing calculated from the proportion of time spent at risk (mobile phone data, travel survey) or the genetic data using a Rand Index (*Rand, 1971*) which is a measure of similarity between two data clusters. For a set of n locations (L) and two clusters (X, Y) of L, the Rand index is calculated as:

$$R = \frac{a+b}{\binom{n}{2}}$$

Where $a$ is the number of pairs of elements in L that are in the same subset in X and Y and $b$ is the number of pairs of elements in L that are in different subsets in X and Y.

## Estimates of cross-border importation and connectivity

To estimate cross-border importation, all genotyped infections with a residence in northeastern Namibia and with no reported international travel were aggregated to the respective district. Individuals who reported international travel or with an international residence were assigned to the destination of the reported travel or the residence location. The majority of these individuals reported either a residence in or travel to locations in the nearest bordering province of southern Angola (n = 219 genotyped cases) and Western Province of Zambia (n = 9, genotyped cases). In addition, previously genotyped infections from different provinces of northern Angola (n = 137) and Southern Province of Zambia (n = 96) were included in the analyses. Cross-border analyses did not include any mobile phone data since available data were limited to cell towers within Namibia. Pairwise proportions of highly related infections were compared between four health districts in Namibia (Rundu, Nyangana, Andara and Zambezi); four locations in Angola (Northern Angola, Calai and Dirico municipalities, and elsewhere in southern Angola) and two locations in Zambia (Western and Southern Provinces). The relative importation estimate between pair of locations was calculated as:

$$\textit{Importation estimates from travel history data}: I_{AB} = T_{AB}\frac{i_A}{i_B} \text{ and } I_{BA} = T_{BA}\frac{i_B}{i_A}$$

$$\textit{Importation estimates from parasite genetic data}: I_{AB} = G\frac{i_A}{i_B} \text{ and } I_{BA} = G\frac{i_B}{i_A}$$

Where $I_{AB}$ and $I_{BA}$ are importation estimates based on genetic data from location A to location B and vice versa; $T_{AB}$ and $T_{BA}$ is the proportion of time at risk for those individuals who reported travel from location A to B and vice versa; $G$ is the proportion of highly related infections between location A and B and $i_A$ and $i_B$ represent malaria incidence from local health system data at locations A and B, respectively.

## Acknowledgments

We thank all study participants and their parents and guardians and all field staff in northeastern Namibia health facilities. We acknowledge the Southern and Central Africa International Centers of Excellence in Malaria Research for the southern Zambia data and the Jiangsu Institute of Parasitic Diseases for the northern Angola data. We would like to thank the Namibia National Vector-borne Diseases Control Programme for their help in procuring the mobile phone data. We are also grateful to MTC Namibia and Ms. Bonita Graupe for sharing these mobile phone data through a written data-sharing agreement and for helping with extractions. BG is a Chan Zuckerberg Biohub investigator.

## Additional information

### Funding

| Funder | Author |
|---|---|
| Bill and Melinda Gates Foundation | Sofonias Tessema<br>Bryan Greenhouse |
| Burroughs Wellcome Fund | Amy Wesolowski |
| National Institutes of Health | Amy Wesolowski |
| Chan Zuckerberg Biohub | Bryan Greenhouse |

The funders had no role in study design, data collection and interpretation, or the decision to submit the work for publication.

### Author contributions

Sofonias Tessema, Data curation, Formal analysis, Validation, Investigation, Visualization, Methodology, Writing—original draft, Writing—review and editing; Amy Wesolowski, Data curation, Formal analysis, Validation, Methodology, Writing—original draft, Writing—review and editing; Anna Chen, Jordan Wilheim, Victor A Alegana, Munyaradzi Tambo, Investigation; Maxwell Murphy, Software, Investigation; Anna-Rosa Mupiri, Investigation, Project administration; Nick W Ruktanonchai, Andrew J Tatem, Bradley Didier, Investigation, Writing—review and editing; Justin M Cohen, Adam Bennett, Hugh JW Sturrock, Roland Gosling, Michelle S Hsiang, David L Smith, Writing—review and editing; Davis R Mumbengegwi, Conceptualization, Field supervision, Investigation, Project administration, Writing—review and editing; Jennifer L Smith, Investigation, Project administration, Writing—review and editing; Bryan Greenhouse, Conceptualization and field supervision

### Author ORCIDs

Sofonias Tessema http://orcid.org/0000-0003-1057-5310
Amy Wesolowski https://orcid.org/0000-0001-6320-3575
Maxwell Murphy http://orcid.org/0000-0003-0332-4388
David L Smith http://orcid.org/0000-0003-4367-3849

### Ethics

Human subjects: Ethical approval for the study was obtained from the Institutional Review Boards of the University of Namibia and the University of California, San Francisco (Identification numbers 15-17422 and 14-14576). Informed consent was obtained from all participants or the parents of all children participated in the Zambezi study. For the Kavango study, IRB approval was obtained but no

informed consent was collected as all samples (used RDTs) and de-identified data were collected during routine surveillance.

## Decision letter and Author response

Decision letter https://doi.org/10.7554/eLife.43510.021
Author response https://doi.org/10.7554/eLife.43510.022

# Additional files

## Supplementary files

• Supplementary file 1. Microsatellites and associated metadata for 2585 infections genotyped in this study.
DOI: https://doi.org/10.7554/eLife.43510.018

• Transparent reporting form
DOI: https://doi.org/10.7554/eLife.43510.019

## Data availability

All data generated or analyzed during this study are included in the manuscript and supplementary files.

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
