## [Decision Letter]

Thank you for submitting your article "Using parasite genetic and human mobility data to infer local and cross-border malaria connectivity in Southern Africa" for consideration by *eLife*. Your article has been reviewed by two peer reviewers, and the evaluation has been overseen by a Reviewing Editor and Neil Ferguson as the Senior Editor. The following individuals involved in review of your submission have agreed to reveal their identity: Samir Bhatt (Reviewer #1) and Thomas S Churcher (Reviewer #2).

The reviewers have discussed the reviews with one another and the Reviewing Editor has drafted this decision to help you prepare a revised submission.

Overall, it was felt there is a need for this type of work, the methods are appropriate and the results are clear, though it was felt a little more detail on the methods and rationale for some of the statements given in the Discussion was needed.

Summary:

The problem of local and cross-border malaria importation is at the forefront of the malaria elimination efforts and previous methods for quantifying this have severe limitations. In this paper the authors develop a framework for incorporating genetic data into malaria surveillance. The authors make use of parasite genetic data combined with estimated patterns of human mobility from mobile phone data in northeastern Namibia. The fine-scale spatial structure in the genetic data indicated that most cases resulted from local transmission. However, combining the genetic data with travel histories provided evidence of parasite connectivity over hundreds of kilometers, both within Namibia and across the borders with Angola and Zambia.

Essential revisions:

1) The Discussion starts by saying that "in this study, we demonstrated that augmentation of traditional malaria surveillance with parasite genetics provided positive, empirical data that the majority of malaria cases observed in northeastern Namibia were due to local transmission". The paper presents lots of data that gives this impression but the authors should state in a following sentence which results specifically gives them this strong conclusion. This would help the reader isolate what is the key information and this is especially needed given the lack of genetic samples from just north of the border. Though the reviewers agree with the statement, some of the language could be tapered.

2) The authors first use published methods to look for a geographical gradient in parasite genetics and both came out negative for substantial geographical clustering so they derive another method. The conclusions should describe why more conventional methods of clustering failed to find a significant result and why the method they develop should be used in future (if that is the authors opinion).

3) Subsection “Evidence of cross-border connectivity between Namibia, Angola and Zambia”, second paragraph. Much is made that the genetics are more similar to southern Angola than the north, though it should be stressed that geographically south Angola and Namibia are much closer geographically. Saying that there is a breakpoint within Angola (as the legend states) seems to be an over-extrapolation of the results, especially give the relatively low sampling in the sites outside of Namibia. Northern Angola data were also collected up to 2 years before, and this should be mentioned.

---

## [Author Response]

Essential revisions:1) The Discussion starts by saying that "in this study, we demonstrated that augmentation of traditional malaria surveillance with parasite genetics provided positive, empirical data that the majority of malaria cases observed in northeastern Namibia were due to local transmission". The paper presents lots of data that gives this impression but the authors should state in a following sentence which results specifically gives them this strong conclusion. This would help the reader isolate what is the key information and this is especially needed given the lack of genetic samples from just north of the border. Though the reviewers agree with the statement, some of the language could be tapered.

We thank the reviewers for pointing out the need to clarify our statement. The distinction we were trying to make was between positive evidence versus negative evidence, i.e. a lack of travel history, and did not mean to imply that we were “positive” this is what the data showed. We have edited language to better explain what aspects of the genetic data support this conclusion and to make the language more clear as a comparison with the (also supportive) information being provided by travel history:

*“*First, parasite genetics provided positive evidence that the majority of malaria cases observed in northeastern Namibia were due to local transmission, evidenced by the strong fine-scale spatial structure in the genetic data. […] This is a key piece of programmatically relevant information that would have been difficult to confirm with negative evidence, i.e. merely based on a lack of history consistent with importation, especially given potential disincentives for individuals to report living outside of Namibia.”

2) The authors first use published methods to look for a geographical gradient in parasite genetics and both came out negative for substantial geographical clustering so they derive another method. The conclusions should describe why more conventional methods of clustering failed to find a significant result and why the method they develop should be used in future (if that is the authors opinion).

We thank the reviewers for highlighting the need to clarify why conventional measures were limited in this study. We have amended the Discussion section to clarify the limitations of classical measures in this study and emphasize the need to develop novel tools to estimate genetic relatedness and importation from genetic data at a fine-scale level:

“In this study site, classical methods for measuring or visualizing genetic differentiation (e.g., Gst, STRUCTURE, or phylogenetic trees) had limited utility due to the marginal differences in allele frequencies between geographically proximal locations in this relatively compact study site and the inability to utilize all information from polyclonal infections. […] Estimation of pairwise genetic relatedness between all infections in this study allowed the incorporation of data from all parasites detected in infections and extraction of useful signals of recent transmission created by recombination and cotransmission of multiple parasites.”

3) Subsection “Evidence of cross-border connectivity between Namibia, Angola and Zambia”, second paragraph. Much is made that the genetics are more similar to southern Angola than the north, though it should be stressed that geographically south Angola and Namibia are much closer geographically. Saying that there is a breakpoint within Angola (as the legend states) seems to be an over-extrapolation of the results, especially give the relatively low sampling in the sites outside of Namibia. Northern Angola data were also collected up to 2 years before, and this should be mentioned.

The reviewers make a good point. The Results section and Figure 5 legend have been updated to highlight the lack of extensive sampling in Angola and the geographic proximity of northern Namibia and southern Angola, including removing the word “breakpoint” as follows:

“Overall, infections from Namibia were on average 142 times more likely to be genetically related to those from southern Angola and 191 times to Zambia than to those from northern Angola, indicating substantial parasite mixing within the geographically connected Namibia-Angola-Zambia regional block. […] Within Namibia, infections from Andara and Nyangana were 3 and 4 times more likely to be genetically related to Zambezi than infections sampled from nearby Rundu, respectively (Figure 5B).”

Figure 5 legend (updated section):

“Northern Angola was not strongly related to samples in the study area and corresponding cross-border region, suggesting that parasite flow between southern and northern Angola is less than that between (more geographically proximate) cross-border regions of northern Namibia and southern Angola.”